# Effect of Free Cross-Linking Rate on the Molding of Bulk SiOC Ceramics

**DOI:** 10.3390/ma16062446

**Published:** 2023-03-19

**Authors:** Lei Zheng, Weilian Sun, Zhijian Ma, Hongchao Ji, Bo Sun

**Affiliations:** 1College of Mechanical and Electrical Engineering, Hebei Agricultural University, Baoding 071201, China; 2College of Mechanical Engineering, North China University of Science and Technology, Tangshan 063210, China; 3Jidong Rizhang Energy-Saving Fan Manufacturing Co., Ltd., Tangshan 063210, China

**Keywords:** polymer-derived ceramics, SiOC ceramics, HPSO/D_4_^Vi^, bulk ceramics, free cross-linking rate

## Abstract

Polymer-derived ceramics (PDCs) have many advantages in ceramic molding and ceramic properties, but because of the obvious volume shrinkage in the process of precursor transformation into ceramics, it is easy for defects to appear in the forming process of bulk PDCs. Herein, theoretical analyses and experimental studies were carried out to improve the quality of sintered samples and realize the parametric design of raw materials. Firstly, based on the HPSO/D_4_^Vi^ cross-linking system, the mathematical model of the free cross-linking ratio was established, and the theoretical value was calculated. After that, the samples with different free cross-linking rates were heated at 450 °C and 650 °C for different holding times. It was found that the free cross-linking ratio (α) had a significant impact on the weight loss of the samples. When the difference of the α value was 10%, the difference of the samples’ weight loss ratio could reach 30%. Finally, the morphology of sintered products with different α values was analyzed, and it was found that obvious defects will occur when the free cross-linking ratio is too high or low; when this value is 40.8%, dense and crack-free bulk ceramics can be obtained. According to analysis of the chemical reaction and cross-linking network density during sintering, the appropriate value of the free cross-linking ratio and reasonable control of the cross-linking network are beneficial for reducing the loss of the main chain element and C element, alleviating the sintering stress, and thus obtaining qualified pressureless sintered bulk ceramic samples.

## 1. Introduction

Compared with traditional ceramic technology, polymer-derived ceramics (PDC) have obvious advantages in composition design and processing technology [1]. PDCs can regulate ceramic products through the design of precursor molecular structures and ceramic transformation conditions. Through the organic–inorganic amorphous–crystalline transformation, the preparation of ceramic materials with special composition and phase structure can be realized, which is difficult to achieve in traditional ceramics [2,3,4]. Moreover, the precursor polymer has obvious physical properties of organic polymers, which can be formed by the molding technology usually applicable to organic polymers [5], such as film coating [6], impregnation [7], vapor deposition [8], electrostatic silk imitation [9], 3D printing [10], etc., and then ceramic products can be obtained by high-temperature pyrolysis. Therefore, PDCs play an important role in the production of high-performance ceramics with complex structures [11], and have important applications in aerospace [12], new energy [13], optical metamaterials [14], military [15], and other fields.

Due to the high density difference between the polymer precursor and ceramic phase, there will be a high volume shrinkage when the polymer precursor is converted to the ceramic phase, which is easy to produce defects and cracks during pyrolysis, affecting the performance of ceramic components [10]. Therefore, PDCs have more advantages in one-dimensional and two-dimensional structure forming, such as ceramic film, ceramic coating, ceramic fiber, porous structure, etc., while in the forming of a three-dimensional block structure, the volume effects of filler or pressure forming are usually used to obtain a higher-strength compact ceramic block structure. For example, Zhang et al. [16] prepared three-dimensional SiC/SiC ceramic components by adding SiC whiskers and SiC particles to polycarbosilane. The research results showed that the mechanical properties of the ceramic materials were significantly improved after adding whiskers and particles to the polymer precursor, and the ceramic sample with a tensile strength of 21.3 MPa and a fracture toughness of 1.89 MPa·m^1/2^ was obtained. With the quantity of added SiC whiskers unchanged, the linear shrinkage of the ceramic materials decreased from 18.32% to 8.3% with the increase in SiC particles. The weight loss decreased from 17.5% to 10.6%, which was mainly due to the low porosity of ceramic materials prepared by the combination of ceramic particles and whiskers, and the ceramic particles enhanced the strength of the samples. Sun et al. [17] used solid ammonium carbamate as the ammonia source, methyl vinyl dichlorosilane and vinyl trichlorosilane as the mixed silicon source to synthesize the SiCNO ceramic precursor, and prepared ceramic blocks by the hot-pressing sintering method. When T < 1500 °C, the product is amorphous SiCNO ceramics. The ceramic density (2.15 g·cm^−3^), bending strength (138.1 MPa), and fracture toughness (2.32 MPa·m^1/2^) were obtained when T = 1500 °C. When T > 1500 °C, the SiCNO material was decomposed and Si2N2O was precipitated from the matrix. Because of the sp2 hybridization of C-N, nitrogen atoms play the role of “bridging nitrogen”, reducing the tendency of carbon precipitation in the materials as free carbon.

Although the pressure condition is effective at improving the performance of ceramic samples, it also limits the forming process of precursor ceramics and restricts the complexity of ceramic parts. Therefore, researchers started from more fundamental factors to study the influence of the precursor composition, proportion, and sintering process on ceramic yield and stress, so as to obtain precursor ceramic blocks with better performance and fewer defects under pressureless conditions [18]. Based on DLP technology, Schmidt et al. [19] mixed high-yield polysiloxane and photosensitive polysiloxane containing acrylate groups into photocuring materials. By adjusting the ratio of the two components, they realized the control of ceramic yield, shrinkage, and other parameters. When the proportion of photosensitive polysiloxane was 50%, the ceramic yield was 40% and the shrinkage was 42%; with the proportion of photosensitive polysiloxane reduced to 33%, the ceramic yield was 60% and the shrinkage was 30%. Yu et al. [20] found that hyperbranched polycarbosilane synthesized from chloromethyl-dichlorosilane, chloromethyl-trichlorosilane, and allyl chloride, with cyclohexanone oxide cobalt naphthenate as catalyst, could realize the cross-linking reaction at room temperature, so the ceramic yield could reach 70%. With the introduction of the alkynyl group, the cross-linking density could be further improved, and the ceramic yield could reach 82.5%.

The designability of precursor molecules is one of the most important characteristics that distinguish precursor ceramics from traditional ceramics [21]. Herein, to better understand the effects of the precursor molecular structure, raw material ratio, and sintering process on sintering stress and ceramic yield, the concept of the free cross-linking rate and its theoretical calculation method were proposed for the two-component precursor system of polymethylhydrosiloxane (HPSO) and polytetramethyltetravinyllic cyclic tetrasililoxane (D_4_^Vi^). The ceramic yield and macro- and micromorphology differences of ceramic samples under different free cross-linking ratios were compared through experiments, which provided a theoretical basis for the molecular design of precursors for the pressureless formation of high-yield compact ceramic blocks.

## 2. Calculation of Free Cross-Linking Rate

In the process of thermal curing, the severity of the cross-linking reaction is not completely consistent with the curing time, and the yield of pyrolytic ceramics is not consistent with the content of active groups. This is caused by the difference in the molecular weight of the precursor polymer. The silicone polymer with lower molecular weight needs to be cross-linked to generate a silicone polymer with higher molecular weight, and then it can be cross-linked into a network structure. This shows that the speed of the cross-linking reaction and the network density of cross-linking products cannot be determined by the content of active groups alone. For this reason, we put forward the concept of the free cross-linking ratio and deduced the calculation method of its theoretical value. The degree of cross-linking reaction can be evaluated by easily measured or known raw material parameters, such as active group content and molecular weight, and parametric design of raw material parameters can be realized.

### 2.1. Modeling of Free Cross-Linking

In order to better characterize the relationship between curing time and the number of cross-linking points in the system, the free cross-linking ratio (α) was defined as the average mass ratio of organics that are occupied by non-cross-linking groups (such as methyl) and cannot be cross-linked. Ideally, it is considered that no self-cross-linking reaction of the vinyl group or dehydrogenation reaction of the hydrosilyl group occur under the conditions of thermal cross-linking reaction. It is assumed that there are three types of cross-linking points: the effective cross-linking point which formed by C=C and Si-H; the cross-linked point which formed on the Si atom and adjacent chained atom (O atom); and the invalid cross-linking point, that the cross-linked point is occupied by nonreactive atoms (such as methyl, redundant vinyl, redundant hydrogen, etc.), which cannot undergo a cross-linking reaction, and it is only a virtual cross-linking point. The total cross-linking point is the Summation of the effective cross-linking point, the cross-linked point, and the invalid cross-linking point. Hence, the parameters are set as follows:$n_{\mathrm{SiV}}$: The number of silicon atoms on a molecular chain (or ring) of vinyl silicone oil;$n_{C=\mathrm{CV}}$: The amount of vinyl on a molecular chain (or ring) of vinyl silicone oil;$n_{\mathrm{CH}3V}$: The number of methyl groups on a molecular chain (or ring) of vinyl silicone oil;$n_{\mathrm{OV}}$: The number of O atoms on a molecular chain (or ring) of vinyl silicone oil;$n_{\mathrm{SiH}}$: The number of silicon atoms on a molecular chain (or ring) of hydrogen silicone oil;$n_{\mathrm{HH}}$: The amount of hydrogen on a molecular chain (or ring) of hydrogen silicone oil;$n_{\mathrm{CH}3H}$: The number of methyl groups on a molecular chain (or ring) of hydrogen silicone oil;$n_{\mathrm{OH}}$: The number of oxygen atoms on a molecular chain (or ring) of hydrogen silicone oil.$\varphi_{C=C}$: Mass content of vinyl$\varphi_{H}$: Mass content of hydrosilyl group


The flow of model and calculation is shown in Figure 1.

Since the oxygen atom on the chain can only extend the chain and cannot provide additional cross-linking points, the total number of cross-linking points of a molecular chain (or ring) of vinyl silicone oil is twice the total number of silicon atoms:$\;2n_{\mathrm{SiV}}$. Similarly, the total number of cross-linking points of a molecular chain (or ring) of hydrogen silicone oil is twice the total number of silicon atoms:$\;2n_{\mathrm{SiH}}$. Therefore, the number of cross-linked points in cyclovinyl silicone oil is$\;n_{\mathrm{SiV}}$.
(1)$$n_{\mathrm{SiV}}=\frac{n_{\mathrm{CH}3V}+n_{C=\mathrm{CV}}}{2},$$

Likewise, the number of cross-linked points in chain hydrogen-containing silicone oil is $\left(n_{\mathrm{SiH}}-1\right)$:(2)$$n_{\mathrm{SiH}}=\frac{n_{\mathrm{CH}3H}+n_{C=\mathrm{CH}}}{2}-1,$$

The cross-linked point and effective cross-linked point can improve the cross-linking degree of the polymer system, while the ineffective cross-linked point cannot improve the cross-linking degree of the cross-linked system. Accordingly, when the number of invalid cross-linking points increases, the number of effective cross-linking points and cross-linked points will decrease, and the cross-linking difficulty of the polymer system or the density of the cross-linking network will decrease.

When there are four invalid cross-linking points on the Si atom, it is impossible to cross-link; when there are three invalid cross-linking points on the Si atom, it can only be cross-linked with adjacent molecules to form larger molecules; when there are two invalid cross-linking points on the silicon atom, theoretically, the molecular chain can only grow infinitely to form a linear structure, and cannot be cross-linked into a network. Only when there are fewer than two invalid cross-linking points on the silicon atom can the cross-linking become a network. Moreover, it is possible to cross-link into a more dense network structure when the average number of invalid cross-link points on each Si atom is less.

### 2.2. Calculation of Invalid Cross-Linking Point

For simple polymers, there is a certain relationship between the content of active groups and the molar weight of each group in the molecule. 

For annular vinyl silicone oil:(3)$$\varphi_{C=C}=\frac{27n_{C=\mathrm{CV}}}{28n_{\mathrm{SiV}}+16n_{\mathrm{SiV}}+15n_{\mathrm{CH}3V}+27n_{C=\mathrm{CV}}},$$

For a hydrogen-containing silicone oil long-chain structure:(4)$$\varphi_{H}=\frac{n_{\mathrm{HH}}}{28n_{\mathrm{SiH}}+16\left(n_{\mathrm{SiH}}-1\right)+15n_{\mathrm{CH}3H}+27n_{C=\mathrm{CH}}},$$

Combined with Formulas (1) and (3), the number of methyl groups on each silicon atom of vinyl silicone oil with an annular structure can be calculated:(5)$$C_{V}=\frac{n_{\mathrm{CH}3V}}{n_{\mathrm{SiV}}}=\frac{54-98\varphi_{C=C}}{27-12\varphi_{C=C}},$$

Similarly, Formulas (2) and (4) can be used to calculate the number of methyl groups on each silicon atom of hydrogen-containing silicone oil with a chain structure:(6)$$C_{H}=\frac{n_{\mathrm{CH}3H}}{n_{\mathrm{SiH}}}=\frac{2-64\varphi_{H}}{14\varphi_{H}+1}+\frac{14\varphi_{H}+2}{n_{\mathrm{SiH}}\left(14\varphi_{H}+1\right)},$$

### 2.3. Calculation of Free Cross-Linking Rate

When the vinyl content and hydrogen content in the solution are 1:1, and it is considered that polymerization reaction continues between vinyl groups or dehydrogenation reaction continues between silyl groups, the active groups of the system are effective cross-linking points. Since every two chemical bonds on the silicon atom are combined into a cross-linked chemical bond, and there are two cross-linked bonds on each silicon atom, the cross-linking vacancy rate $\alpha_{V}$ and $\alpha_{H}$ satisfies the following relations:

Vinyl silicone oil:(7)$$\alpha_{V}=\frac{\frac{C_{V}}{2}}{2}=\frac{C_{V}}{4},$$

Hydrogen silicone oil:(8)$$\alpha_{H}=\frac{\frac{\alpha_{H}}{2}}{2}=\frac{\alpha_{H}}{4},$$

Then, the proportion α of the total equivalent void cross-linking points in the polymer solution is α:(9)$$\alpha =\frac{\alpha_{V}m_{V}+\alpha_{H}m_{H}}{m_{V}+m_{H}},$$
where $m_{V}$ is the weight of vinyl silicone oil, and $m_{H}$ is the weight of hydrogen silicone oil.

Because the molar weight of vinyl silicone oil and hydrogen-containing silicone oil is the same, their weight also has a corresponding relationship: $m_{V}=27m_{H}$; then, the molecular weight ratio is substituted:(10)$$\alpha =\frac{27\alpha_{V}\varphi_{H}+\alpha_{H}\varphi_{C=C}}{27\varphi_{H}+\varphi_{C=C}},$$

In the formula, $\varphi_{C=C}$, $\varphi_{H}$, $n_{\mathrm{SiV}}$, and $n_{\mathrm{SiH}}$ can be obtained through raw material parameters, spectrum detection, chemical titration, viscosity test, etc.$;{\;\alpha }_{V}$ and $\alpha_{H}$ can be obtained by Formulas (7) and (8), all of which are known parameters. When α < 0.5, the average number of invalid cross-linking points on each silicon atom is less than 2, meaning that the polymer can cross-link the network polymer; when α > 0.5, the number of invalid cross-linking points on each silicon atom is more than 2 on average, then the polymer can only be cross-linked into linear or local network cross-linking, which will lead to serious weight loss during sintering, and a dense ceramic structure cannot be obtained.

## 3. Materials and Methods

### 3.1. Main Raw Materials

Tetramethyltetravinyl cyclotetrasiloxane (D_4_^vi^, molecular weight 344, active group content $\varphi_{C=C}=31\%$) was provided by Guangzhou Shuangtao Fine Chemical Co., Ltd. Polymethylhydrosiloxane (HPSO, Guangzhou Shuangtao Fine Chemical Co., Ltd., Guangzhou, China): Sample A, viscosity is 22 mPas, active H group content is $\varphi_{H}=0.2\%$, molecular weight is 1000; Sample B, viscosity is 200 mPas, active H group content is $\varphi_{H}=0.2\%$, molecular weight is 15,000; Sample C: viscosity is 200 mPas, active H group content is $\varphi_{H}=0.3\%$, molecular weight is 15,000; Sample D: viscosity is 200 mPas, active H group content is $\varphi_{H}=0.5\%$, molecular weight is 15,000. Catalyst: Karstedt catalyst, Shenzhen Aokai Organosilicon Co., Ltd. (Shenzhen, China), with an effective ingredient concentration of 3000 ppm. Calcined kaolin, 8000 mesh (particle diameter less than 1.6 μm), with a two-dimensional lamellar structure was purchased from Hebei Jijiang Technology Co., Ltd. (Shijiazhuang, China).

### 3.2. Experimental Methods

Hydrogen-containing silicone oil and vinyl silicone oil were prepared according to the active group ratio of 1:1. After stirring for 10 min with magnetic agitators, Karstedt catalyst was added at the dosage of 20 ppm. After stirring for 30 min, samples A, B, C, and D were obtained: A (*α* = 46.7%), B (*α* = 43.3%), C (*α* = 40.8%), D (*α* = 36.6%). The specific experimental steps are as follows:

(1) The four samples A, B, C, and D were taken, respectively, each of which was 20 g. The samples were solidified in an incubator at 80 °C and kept for 1 h. After that, the samples were heated to 150 °C at 2 °C/min in a tubular-atmosphere furnace for 1 h, and then heated to 450 °C at 2 °C/min and kept for 0.5 h, 1 h, 2 h, and 4 h, respectively. The temperature was lowered below 100 °C at the rate of 2 °C/min and cooled naturally to obtain samples E11~E14 (holding for 0.5 h), F11~F14 (holding for 1 h), G11~G14 (holding for 2 h), and H11~H14 (holding for 4 h).

(2) The four samples A, B, C, and D were taken, respectively, weighing 20 g. The samples were heated and solidified for 1 h in a constant-temperature oven at 80 °C, then heated to 150 °C for 1 h in a tubular-atmosphere furnace at a rate of 2 °C/min. Subsequently, they were heated to 700 °C at the rate of 2 °C/min for 0.5 h, 1 h, 2 h, and 4 h, respectively. The samples were cooled down below 100 °C at the rate of 2 °C/min, and were naturally cooled to obtain E21~E24 (holding for 0.5 h), F21~F24 (holding for 1 h), G21~G24 (holding for 2 h), and H21~H24 (holding for 4 h).

(3) The 8000-mesh calcined kaolin was dried in a constant-temperature drying oven for 12 h, and then the ball mill was used for 2 h to obtain the white and evenly dried kaolin powder. A total of 61 g of calcined kaolin was put into four beakers and 50 g of samples A, B, C, and D were added, respectively. After full stirring, a white slurry with good fluidity was obtained by standing at 0 °C for 4 h. The samples were stirred by a rotary mixer at 1000 rpm for 15 min, placed at 0 °C for 4 h, and stirred for 5 min in a vacuum vibrating agitator. The samples were placed in an incubator at 80 °C for 1 h after curing, and heated to 1000 °C in a tubular-atmosphere furnace at a heating rate of 2 °C/min. The samples were kept for 1 h at 150 °C, 400 °C, 620 °C, and 1000 °C respectively, then dropped below 100 °C at the rate of 2 °C/min and cooled naturally. The samples were cut into 10 mm × 10 mm × 40 mm to obtain E3, F3, G3, and H3.

### 3.3. Experimental Characterization

The weight of samples E11~E14, F11~F14, G11~G14, and H11~H14 were recorded before and after sintering, and the weight loss of the samples was calculated. The same procedure also applied to samples E21~E24, F21~F24, G21~G24, and H21~H24. The sintered macroscopic morphologies of the samples E3, F3, G3, and H3 were observed, and the samples were fixed on the sample table with conductive adhesive for gold-spraying treatment, and then the micromorphologies of the cracking products were observed under S-4800 scanning electron microscope, which is made by Hitachi, LTD. (Tokyo, Japan).

## 4. Results and Discussion

### 4.1. Weight Loss of Samples with Different Free Cross-Linking Rates under 450 °C

Table 1 shows the weight loss ratio of samples (E11~H14) at 450 °C under different holding periods. As a whole, the weight loss ratio decreased from the bottom left to the top right.

The data in longitudinal comparison of the table showed that the holding time at 450 °C had a significant impact on the weight loss ratio. According to the thermal weight loss characteristics of siloxane precursor system [22,23], the weight loss before 450 °C was mainly divided into two stages: first, the weight loss of the system caused by the escape of non-cross-linked small molecules, and then the separation of small molecules generated in the cracking process. However, the weight loss difference of sample E1 group at 0.5 h and 4 h was more than 25%, indicating that not only did CH4 small-molecule pyrolysis gas escape, but also the loss of Si atoms during pyrolysis. According to the spectral characteristics of precursors of SiOC system [24,25], the remaining chemical bonds in the system after 350 °C were mainly Si-C bonds, Si-O bonds, C-H bonds, C-C bonds, and a small amount of unstable Si-Si bonds. At 450 °C, the activation energy was above 200 kJ/mol, close to the activation energy required for Si-C bond and C-C bond, which was 263 kJ/mol and 234 kJ/mol. The C-H bond and Si-O bond with stronger bond energy did not reach the breaking temperature [26], while the Si-C bond and C-C bond were destroyed, generating a large number of free radicals, and the following reactions occurred:(11)$$\equiv \mathrm{Si}-{\mathrm{CH}}_{3}\;\to \;\equiv \mathrm{Si}\cdot +\cdot {\mathrm{CH}}_{3}$$
(12)$$\equiv \mathrm{Si}-{\mathrm{CH}}_{2}-{\mathrm{CH}}_{2}-\mathrm{Si}\equiv \;\to \;\equiv \mathrm{Si}-{\mathrm{CH}}_{2}\cdot +\cdot {\mathrm{CH}}_{2}-\mathrm{Si}\equiv$$
(13)$$\equiv \mathrm{Si}-{\mathrm{CH}}_{2}-\to \;\equiv \mathrm{Si}\cdot +\cdot {\mathrm{CH}}_{2}-$$

The free radicals formed in the above reaction could seize other groups or H atoms from the main chain, and could also combine with each other, thus forming a variety of possible results, such as improving the density of the cross-linked network, reducing the density of the cross-linked network, or escape in small molecular form. It could be expressed as the following reaction.

Reactions that increased the density of cross-linked networks:(14)$$\equiv \mathrm{Si}\cdot +\cdot {\mathrm{CH}}_{2}-\mathrm{Si}\equiv \;\to \;\equiv \mathrm{Si}-{\mathrm{CH}}_{2}-\mathrm{Si}\equiv$$
(15)$$\equiv \mathrm{Si}-{\mathrm{CH}}_{2}\cdot +\;{\cdot \mathrm{CH}}_{2}-\mathrm{Si}\equiv \;\to \;\equiv \mathrm{Si}-{\mathrm{CH}}_{2}-{\mathrm{CH}}_{2}-\mathrm{Si}\equiv$$
(16)$$\equiv \mathrm{Si}-{\mathrm{CH}}_{2}\cdot +{\;\mathrm{CH}}_{3}-\mathrm{Si}\equiv \;\to \;\equiv \mathrm{Si}-{\mathrm{CH}}_{2}-{\mathrm{CH}}_{2}-\mathrm{Si}\equiv +\;H$$
(17)$$\equiv \mathrm{Si}\cdot +{\;\mathrm{CH}}_{3}-\mathrm{Si}\equiv \;\to \;\equiv \mathrm{Si}-{\mathrm{CH}}_{2}-\mathrm{Si}\equiv +\;H$$

Reactions that decreased the density of cross-linked networks:(18)$$\equiv \mathrm{Si}-{\mathrm{CH}}_{2}-\mathrm{Si}\equiv \;\to \;\equiv \mathrm{Si}-{\mathrm{CH}}_{2}\cdot +\cdot \mathrm{Si}\equiv$$
(19)$$\equiv \mathrm{Si}-{\mathrm{CH}}_{2}\cdot +\cdot {\mathrm{CH}}_{3}\;\to \;\equiv \mathrm{Si}-{\mathrm{CH}}_{2}-{\mathrm{CH}}_{3}$$
(20)$$\equiv \mathrm{Si}\cdot +\cdot {\mathrm{CH}}_{3}\;\to \;\equiv \mathrm{Si}-{\mathrm{CH}}_{3}$$

Reaction to produced small-molecule gas:(21)$${\mathrm{CH}}_{3}\cdot +\cdot {\mathrm{CH}}_{3}\;\to {\;\mathrm{CH}}_{3}-{\mathrm{CH}}_{3}↑$$
(22)$${\mathrm{CH}}_{3}\cdot +\cdot H\;\to {\;\mathrm{CH}}_{4}↑$$
(23)$$H\cdot +\cdot H\;\to {\;H}_{2}↑$$

Therefore, the reaction at this stage became very complex, with multiple possibilities and high repeatability. The local cross-linking network density changed dynamically; however, it could be confirmed that the escape of small molecules is the combination of two or more free radicals, meaning that when small molecules escape, free radicals that could be connected with each other are left on the main chain, which could be recombined into cross-linked bonds in the dynamic equilibrium, such as:(24)$$\equiv \mathrm{Si}-{\mathrm{CH}}_{2}-R_{1}\;+R_{2}-\mathrm{Si}\equiv \;\to \;\equiv \mathrm{Si}-{\mathrm{CH}}_{2}-\mathrm{Si}\equiv \;+R_{1}-R_{2}↑$$

At this time, the Si-C-Si chain was formed between Si atoms originally separated on two molecular chains and cross-linked together, that is, under high-temperature conditions, methyl that had no cross-linking activity provided cross-linking. By removing two methyl groups or one methyl group and one hydrogen atom, a cross-linking structure was established, increasing the number of cross-linking bonds in the system and forming a denser cross-linking network structure, thus effectively inhibiting the loss of Si-O elements, which was shown in a nonlinear relationship between the weight loss ratio and the insulation time. As the reaction proceeds, the density of the cross-linking network increased, reducing the reaction conditions that can lose weight and slowing down the weight loss speed.

Comparing the data in Table 1 horizontally, it was found that the weight loss decreased significantly with the decrease in *α*. With the same content of active H, the precursor polymer with low residual cross-linking ratio had less weight loss, while the weight loss of the precursor polymer with high residual cross-linking ratio increased significantly.

The active group promotion of the sample E1 and F1 groups was completely consistent, but their weight loss ratio was obviously different, which was caused by two reasons. On the one hand, there was a significant difference in the molecular weight of hydrogen-containing silicone oil in the sample E1 and F1 groups. According to the principle of the α value calculation, 50% of the chemical bonds of Si atoms were used to make the system cross-linked into linear macromolecules. On this theoretical basis, 46.7% of the Si-O backbone in the sample E1 group was connected with inactive groups, only 3.3% could make the Si-O backbone cross-linked into a network structure, and 6.7% of the sample F1 group could be used for the Si-O backbone cross-linked into a network structure. For the density of the cross-linking network, the sample E1 group was significantly lower than the sample F1 group. In the cross-linking process, when the uniformity and completeness could not be absolutely provided, there would be a large number of components that were not cross-linked or only cross-linked into macromolecules, which would vaporize and escape during the heating process, resulting in an obvious weight loss. However, a denser network structure of the sample F1 group was formed during the cross-linking process, the requirements for uniformity and completeness in cross-linking reaction were reduced, and the weight loss caused by volatilization was relatively reduced. For the sample G1 and H1 groups, the density of the cross-linking network was much higher than that of the sample E1 and F1 groups, hence only a small amount of non-cross-linked gas escaped during the heating process. On the other hand, the molecular weight of the sample E1 group was significantly lower than that of the sample F1 group, indicating that the number of Si-C bonds formed on each hydrogen-containing silicone oil molecule was significantly less than that of the sample F1 group. It was known that the Si-C bond energy was lower than the Si-O bond energy, and it was more likely to be break at high temperatures. When the Si-C bond on the hydrogen-containing silicone oil was completely broken, it would escape in the form of gas. For the sample F1 group, since there were more Si-O bonds on each molecule on average, the loss of the Si element was lower than that of the sample E1 group. For the sample G1 and H1 groups, the molecular weight was larger and the content of active groups was higher, making the molecular chains less likely to be separated, and their weight loss was mainly small-molecule cracked gases such as methane, ethane and ethylene; consequently, their weightlessness was lower.

According to the comparison of horizontal data and vertical data in Table 1, the weight loss near 450 °C mainly included the escape of unconnected small-molecule gas, the loss of main chain caused by the breakage of cross-linked bond, and the escape of small-molecule pyrolysis gas. The loss of elements on the main chain of polymer was the main reason for the difference in weight loss. The lower free cross-linking ratio and higher density of the cross-linking network were important factors to inhibit the weight loss of the main chain in the system. At the same time, the loss of the main chain would be suppressed with the increase in the density of the cross-linking network. However, due to the large number of methyl groups in the system, the weight loss of small-molecule pyrolysis gas would be sustained under this reaction condition.

### 4.2. Weight Loss of Samples with Different Free Cross-Linking Rates under 700 °C

Table 2 shows the weight loss of samples (E21~H24) at 700 °C for different holding duration. The horizontal comparison table shows that the weight loss ratio of the sample was affected evidently by the *α* value, which was due to the weight loss caused by 400~600 °C during the heating process. However, compared with the data in Table 2, it was found that the weight loss ratio increased slowly with the temperature holding time, indicating that the system gradually turned to the thermal stability stage, but there were still some differences between samples with different free cross-linking rates.

According to the calculation of reaction kinetics [26], the activation energy at this stage was more than 350 kJ/mol, close to the C-H bond breaking condition (387 kJ/mol), and a large number of H radicals were released from the system:(25)≡C−H → ≡C·+·H

The reaction activity of the system was further enhanced by H free radicals, which could combine with each other to synthesize H_2_ and escape, such as:(26)$$H\cdot +\cdot H\to {\;H}_{2}↑$$

Moreover, H radicals could attack the radicals connected to C to form H_2_, such as:(27)$$\equiv \mathrm{CH}\;+\cdot H\;\to \;\equiv C\cdot +{\;H}_{2}↑$$

In addition, H radicals could replace the—CH_3_ group, such as:(28)$$\equiv C-{\mathrm{CH}}_{3}+\cdot H\;\to \;\equiv C\cdot +{\;\mathrm{CH}}_{4}↑\;$$
(29)$$\equiv \mathrm{Si}-{\mathrm{CH}}_{3}+\cdot H\;\to \;\equiv \mathrm{Si}\cdot +{\;\mathrm{CH}}_{4}↑$$

Furthermore, they might also attack other covalent bonds, such as the C-C bond and Si-C bond, then combine into small-molecule gas and escape, leaving a large number of free radicals, such as:(30)$$\equiv C-{\mathrm{CH}}_{3}+\cdot H\;\to \;\equiv \mathrm{CH}\cdot +\cdot {\mathrm{CH}}_{3}$$
(31)$${\mathrm{CH}}_{3}\cdot +\cdot {\mathrm{CH}}_{3}\to {\;\mathrm{CH}}_{3}-{\mathrm{CH}}_{3}↑\;$$

This made the reaction of the system very complex. Although there was a certain mass loss due to the separation of the main chain and the C element in the early stage, and the density of the cross-linking network of the system increased significantly, a large number of H atoms were still preserved in the system in the form of C-H bonds. The weight loss at this stage was mainly caused by the loss of the H atoms, which was due to the stronger bond energy of the H-H bond and the steric hindrance effect. H atoms had fewer chances to attack the Si-C chain and C-C chain; they mainly combined with H on the C atom and produced H_2_ to escape, resulting in a small amount of weight loss and a dense cross-linking network.

With the addition of the C-H bond to the main reaction, each C-H bond provided an additional cross-linking point, greatly changing the number of cross-linkable points of the original cross-linking system. When the reaction was completed and the H atom was completely separated, only Si, O, and C elements remained in the system, forming a highly dense network system, that is, the amorphous [SiOC] structure. In this process, excessive C atoms can combine into graphite form with other excessive C atoms, while more excessive C atoms can combine with Si, O, and C atoms to form the amorphous [SiOC] structure. This is also the reason why the excessive C in the system exists in more than one forms.

At this time, excessive C in the system might form a stable graphite structure during rearrangement, but it was easier to combine the amorphous structure in the [SiOC] system in the form of the C-C bond, Si-C bond, etc. This is highly consistent with the views in a large amount of the literature.

Therefore, the weight loss stage near 700 °C was an important stage in which the system was transformed from organic matter to inorganic matter. The physical properties at this stage also changed significantly, with a large increase in hardness and density and a significant shrinkage in volume, reflecting obvious characteristics of ceramic materials. On the other hand, it was also a process of volume shrinkage and hardness improvement. When the free cross-linking ratio was too low, it was easy to have large stress and structural defects.

Combined with the weight loss performance and the chemical reaction process of the two stages, it could be inferred that the process and significance of the chemical reaction at the 450 °C and 700 °C stage were obviously different: The 450 °C stage was dominated by the fracture and rearrangement of the Si-C bond and C-C bond, which caused obvious weight loss. When the α value was low, it was mainly the loss of the C element; on the contrary, it was the loss of the Si and C elements. When the value was determined, the weight loss ratio had a certain relationship with the heating time, which was manifested as the characteristics of organic matter. The main significance of the reaction process was to establish a high-density Si-O-C-H network to ensure a more stable structure of the system during the continuous heating. For block structures, the high free cross-linking rate and the slow heating rate would reduce the density of the cross-linking network in the system, which would easily lead to an obvious increase in weight loss, a decrease in density, shrinkage deformation, and crack defects. In the 700 °C stage, the C-H bond fracture and rearrangement were the main causes of weight loss, which was mainly caused by the loss of H atoms. The weight loss ratio had a high correlation with the content of H atoms in the system, but a low correlation with the heating time, and the materials showed inorganic properties. The significance of this stage was to establish a Si-O-C network and form an amorphous structure of [Si-O-C]. For block structures, the low free cross-linking rate and the fast heating rate would lead to strong system shrinkage stress and crack defects.

### 4.3. Effect of Free Cross-Linking Rate on Block Structure Molding

Figure 2 shows the macromorphology of the sintered samples E3, F3, G3, and H3. From the conclusion of the previous section, the α value and the weight loss ratio of the sample had the relationship of E3 > F3 > G3 > H3. From the macromorphology, the samples were quite different. The samples with higher α values, such as E3 and F3, had obvious cracks. Sample E3 had obvious morphological deformation; even though sample F3 had no obvious deformation, there were three obvious large cracks and a large number of tiny cracks. Meanwhile, samples G3 and H3 with lower α values retained a relatively complete appearance. No cracks were found in sample G3, and only two obvious cracks appeared in sample H3 at the concentrated stress at the boundary of the sample.

From the microstructure perspective, as shown in Figure 3, there were large differences in sample morphology. The section flatness of samples E3 and F3 with higher α values was poor, indicating that the sample structure was loose and there was a significant difference between the strength of the sintered precursor and the reinforcement particles, and a large number of microcracks were obviously found in the microstructure of sample E3. The defect degree of sample F3 was reduced, but a certain number of microcracks could still be found. Samples G3 and H3 with lower α values had smoother cross sections and more uniform microstructures, with smaller pore sizes and no microcracks.

The macro- and micromorphology studies showed that the α value plays an important role in the properties of sintered samples. During the heating process of the precursor polymer, due to the enhanced molecular vibration under high-temperature conditions, the phenomenon of molecular chain slip was intensified, and the polymer showed strong creep performance and high elastic modulus, which was the basis for the precursor ceramic slurry to remain dense during the shrinkage process. The creep properties and elastic modulus of polymers were affected by heating temperature and the density of the cross-linking networks. When the *α* value was high, it caused obvious weight loss, which reduced the strength of the precursor and increased the shrinkage rate, leaving a large number of pores and cracks in the system and seriously affecting the macrostructure of the sample. When the *α* value was low, the temperature required for creep increased. When the precursor polymer shrunk significantly and a large amount of gas escaped, the sample did not have better deformation capacity and elastic modulus, and brittle fracture cracks would occur.

In the process of converting the precursor polymer into a ceramic, the stress on the object is complex, including the stress caused by volume shrinkage due to weight loss, the stress released due to high-temperature creep, and so on. These factors influence each other complexly. For the HPSO/D_4_^Vi^ system, the free cross-linking rate of 40% is a balance point. In this condition, according to the regular of crosslinking network density, dense precursor ceramic blocks can be obtained by keeping the heating preservation at 400 °C and 600 °C. 

## 5. Conclusions

In this paper, a model of a free cross-linking rate was established. Aiming at the HPSO/D_4_^Vi^ system, a theoretical method was proposed to calculate the free cross-linking rate by molecular weight and active group content. In addition, the weightlessness behavior at different pyrolysis temperatures was studied. Under a condition of 450 °C, the weight loss behavior of the cracking was dominated by C weight loss. It was found that the high free cross-linking rate would lead to the insufficient density of the cross-linking network of the system, resulting in an increase in the weight loss proportion of C, obvious weight loss, and the easy generation of large deformation and stress, eventually inducing structural defects. When cracking at 650 °C, the weight loss behavior dominated by H loss found that the value of the free cross-linking ratio and the holding time would not have a significant impact on the weight loss of the H element, but would significantly increase the density of the cross-linking network. The proper reduction in the value of the free cross-linking ratio and proper heat preservation would help to avoid the excessive density of the cross-linking network, releasing the stress of the sample through gentle creep behavior and reducing the sintering defects. When the free cross-linking rate was 40%, compact crack-free block ceramics were obtained at 400 °C and 600 °C, while the samples with too high or too low free cross-linking rates showed obvious defects.

## Figures and Tables

**Figure 1 materials-16-02446-f001:**
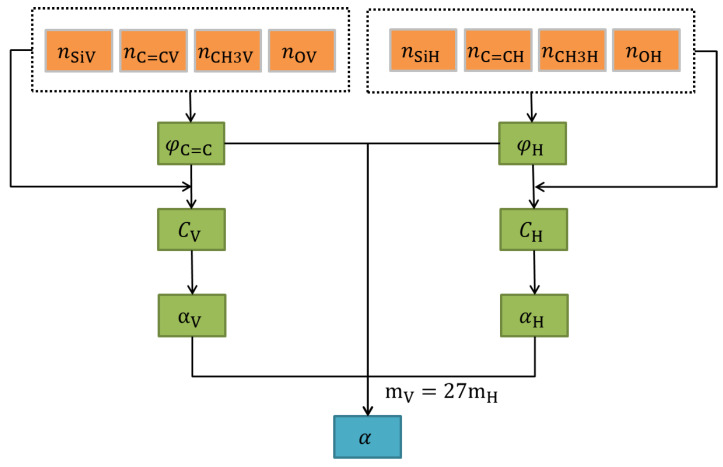
Flow of model and calculation of *α* value.

**Figure 2 materials-16-02446-f002:**
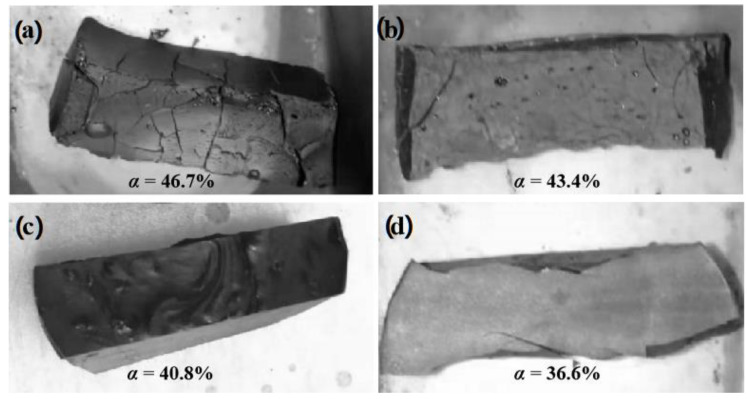
Morphology of sintered samples with different *α* values.

**Figure 3 materials-16-02446-f003:**
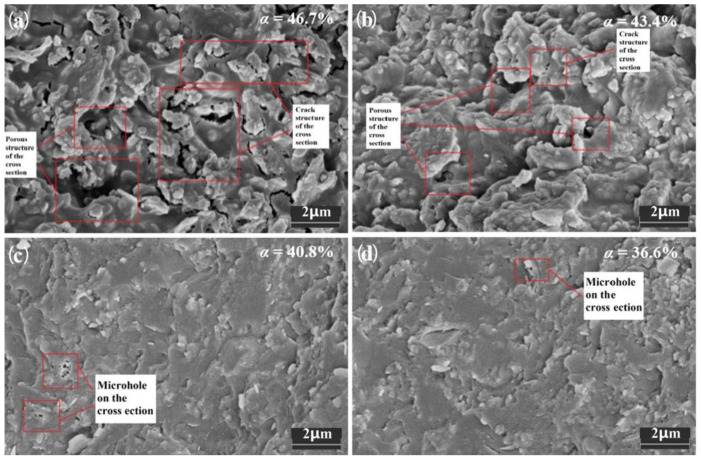
Microstructure of sintered samples with different *α* values.

**Table 1 materials-16-02446-t001:** Weight loss of sample 1 group under different holding times at 450 °C.

Holding Time/h	Weight Loss of Sample E1/%	Weight Loss of Sample F1/%	Weight Loss of Sample G1/%	Weight Loss of Sample H1/%
0.5	26	21	16	12
1	35	28	20	15
2	43	32	23	19
4	52	38	25	20

**Table 2 materials-16-02446-t002:** Weight loss of sample 2 group under different holding times at 700 °C.

Holding Time/h	Weight Loss of Sample E2/%	Weight Loss of Sample F2/%	Weight Loss of Sample G2/%	Weight Loss of Sample H2/%
0.5	47	33	23	19
1	49	34	24	20
2	50	35	24	20
4	52	35	24	20

## Data Availability

Not applicable.

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
