# Peer review of "Effect of Free Cross-Linking Rate on the Molding of Bulk SiOC Ceramics"

_materials, 2023, doi:10.3390/ma16062446_

Round 1
Reviewer 1 Report
I found this article interesting for the readers of the Jornal. However, several parts of the article can be improved.
Abstract . What was the sense to pt 9.%? Maybe, just say~10%.
Chapter 2. I would suggest changing the title of the chapter as follows: Calculation of free-cross-linking rate. No theory in this chapter at all. Just a few simple equations.
Figure 1. What is the sense of this figure? I would suggest removing it. If not, provide an appropriate reference of the code, which was used to generate the casal distribution of the atoms.
Figure 3. Please put a scale inside each panel. This is typical for SEM images.
The text should be checked with the language carrier.
Line 440-444:
During the ceramic process of making precursor polymer,......(word making is missed)
The article requires minor revision and language spelling
Author Response
Response to Reviewer 1 Comments
Effect of Free Cross-linking Rate on the Molding of Bulk SiOC Ceramics
Dear editors and reviewers:
Thank you for your Review Report at 7 March 2023. We have carefully reviewed the comments and have revised the manuscript accordingly. Our responses are given in a point-by-point manner below. Changes to the manuscript are shown in red.
We hope the revised version is now suitable for publication and look forward to hearing from you in due course.
Sincerely,
Lei Zheng
Response to Reviewer: Thank you for your review of our paper. We have answered each of your points below.
Point 1: Abstract . What was the sense to pt 9.%? Maybe, just say~10%.
Response 1: Thank you for your suggestion. The pt 9.% in the abstract is my spelling mistake. I corrected it in my manuscript. It corresponds to the α value difference between sample A and sample D. It should be changed to 10.1% or 10%. I think it is more intuitive to change it to 10% as your suggest.
Point 2: Chapter 2. I would suggest changing the title of the chapter as follows: Calculation of free-cross-linking rate. No theory in this chapter at all. Just a few simple equations.
Response 2: Thank you for your suggestion and your suggestion is very accurate. I revised this title to “Calculation of free-cross-linking rate” in my manuscript. It is a mistake in my expression. This part only introduces the calculation method of the α value, which really cannot be a theory.
Point 3: Figure 1. What is the sense of this figure? I would suggest removing it. If not, provide an appropriate reference of the code, which was used to generate the casal distribution of the atoms.
Response 3: Thank you for your suggestion. This picture really doesn't mean much. So I deleted it in my manuscript and described the phenomenon in words. This phenomenon was discovered 10 years ago. It's just that no one has explained and analyzed it in this point of view. The meaning of the picture is to verify the correctness of the method, however the phenomenon in the picture can be described completely in words.
Point 4: Figure 3. Please put a scale inside each panel. This is typical for SEM images.
Response 4: Thank you for your opinion. This is really my mistake, I forgot to put a scale in each pane, I revised all the SEM images in my manuscript.
Point 5: The text should be checked with the language carrier.
Line 440-444:
During the ceramic process of making precursor polymer,......(word making is missed)
The article requires minor revision and language spelling
Response 5: Thank you for your opinion. I modified the expression of this paragraph in my manuscript.
“In the process of converting precursor polymer into ceramic, the stress on the object is complex, including the stress caused by volume shrinkage due to weight loss, and the stress releasing due to high temperature creep, and so on. These factors influence each other complexly. For the HPSO/D4Vi system, the free cross-linking rate of 40% is a balance point. In this condition, we use the regular pattern of the density of cross-linked networks, keep heating preservation at 400 ℃ and 600 ℃ properly, a dense precursor ceramic block could be obtained.”
I also revised and marked other words in the manuscript. In addition, I added an illustration of the calculation process.
This revision is the consensus of all authors. We tried our best to improve the manuscript and made some changes marked in red in revised paper which will not influence the content and framework of the paper. We appreciate for Editors/Reviewers’ warm work earnestly, and hope the correction will meet with approval. Once again, thank you very much for your comments and suggestions.
We look forward to hearing from you regarding our submission. We would be glad to respond to any further questions and comments that you may have.
I have attached the revised comments to the back of the manuscript and uploaded them as an attachment.
Reviewer 2 Report
In the article by Lei Zheng et al., a model of the speed of free stitching was created. For the HPSO/D4vid system, a theoretical method was proposed for calculating the free crosslinking rate by molecular weight and the content of active groups. The material is written in scientific language. The conclusions correspond to the content. Methods and approaches are described adequately. From the comments, it can be noted that the style of presentation corresponds to the solution of the engineering problem. It is worth adding graphical dependencies and justification for choosing one or another direction of research.
Author Response
Response to Reviewer 2 Comments
Effect of Free Cross-linking Rate on the Molding of Bulk SiOC Ceramics
Dear editors and reviewers:
Thank you for your Review Report at 7 March 2023. We have carefully reviewed the comments and have revised the manuscript accordingly. Our responses are given in a point-by-point manner below. Changes to the manuscript are shown in red.
Sincerely,
Lei Zheng
Point 1: In the article by Lei Zheng et al., a model of the speed of free stitching was created. For the HPSO/D4vid system, a theoretical method was proposed for calculating the free crosslinking rate by molecular weight and the content of active groups. The material is written in scientific language. The conclusions correspond to the content. Methods and approaches are described adequately. From the comments, it can be noted that the style of presentation corresponds to the solution of the engineering problem. It is worth adding graphical dependencies and justification for choosing one or another direction of research.
Response 1: Thank you very much for recognizing and supporting our article. And thank you for your encouragement. We revised the words and expressions of the article to make it more clear to express our views, and we also added an illustration of the calculation process. In addition, we made the changes marked in red in revised paper which will not influence the content and framework of the paper. We appreciate for Editors/Reviewers’ warm work earnestly, and hope the correction will meet with approval. Once again, thank you very much for your comments and suggestions.
We look forward to hearing from you regarding our submission. We would be glad to respond to any further questions and comments that you may have.
I have attached the revised comments to the back of the manuscript and uploaded them as an attachment.